# Chronic sequelae complicate convalescence from both dengue and acute viral respiratory illness

Shirin Kalimuddin[1,2]*, Yii Ean Teh[1], Liang En Wee[1], Shay Paintal[3], Ram Sasisekharan[4,5], Jenny G. Low[1,2,6], Sujata K. Sheth[7], Eng Eong Ooi[2,6,8]*

**1** Department of Infectious Diseases, Singapore General Hospital, Singapore, **2** Program in Emerging Infectious Diseases, Duke-NUS Medical School, Singapore, **3** Accio Systems Pte Ltd, Singapore, **4** Antimicrobial Resistance Interdisciplinary Research Group, Singapore-MIT Alliance for Research and Technology, Singapore, **5** Department of Biological Engineering, Massachusetts Institute of Technology, Boston, Massachusetts, United States of America, **6** Viral Research and Experimental Medicine Centre, SingHealth Duke-NUS Academic Medical Centre, Singapore, **7** Department of Emergency Medicine, Changi General Hospital, Singapore, **8** Saw Swee Hock School of Public Health, National University of Singapore, Singapore

* shirin.kalimuddin@singhealth.com.sg (S.K.); engeong.ooi@duke-nus.edu.sg (E.E.O.)

**Data Availability Statement:** Data cannot be shared publicly because of institutional data regulations on confidential data sharing. Data are available from the Singapore General Hospital

## Abstract

Long Covid has raised awareness of the potentially disabling chronic sequelae that afflicts patients after acute viral infection. Similar syndromes of post-infectious sequelae have also been observed after other viral infections such as dengue, but their true prevalence and functional impact remain poorly defined. We prospectively enrolled 209 patients with acute dengue (n = 48; one with severe dengue) and other acute viral respiratory infections (ARI) (n = 161), and followed them up for chronic sequelae up to one year post-enrolment, prior to the onset of the Covid-19 pandemic. Baseline demographics and co-morbidities were balanced between both groups except for gender, with more males in the dengue cohort (63% vs 29%, p<0.001). Except for the first visit, data on symptoms were collected remotely using a purpose-built mobile phone application. Mental health outcomes were evaluated using the validated SF-12v2 Health Survey. Almost all patients (95.8% of dengue and 94.4% of ARI patients) experienced at least one symptom of fatigue, somnolence, headache, concentration impairment or memory impairment within the first week of enrolment. Amongst patients with at least 3-months of follow-up, 18.0% in the dengue cohort and 14.6% in the ARI cohort experienced persistent symptoms. The median month-3 SF-12v2 Mental Component Summary Score was lower in patients who remained symptomatic at 3 months and beyond, compared to those whose symptoms fully resolved (47.7 vs. 56.0, p<0.001), indicating that patients who self-reported persistence of symptoms also experienced functionally worse mental health. No statistically significant difference in age, gender distribution or hospitalisation status was observed between those with and without chronic sequelae. Our findings reveal an under-appreciated burden of post-infection chronic sequelae in dengue and ARI patients. They call for studies to define the pathophysiology of this condition, and determine the efficacy of both vaccines as well as antiviral drugs in preventing such sequelae.

Research office (contact via email: research. office@sgh.com.sg) for researchers who meet the criteria for access to confidential data.

**Funding:** This work was funded by a SingHealth Foundation Research Grant (SHF/FG622P/2016) awarded to YET, and the National Research Foundation through the Singapore MIT Alliance for Research and Technology Antimicrobial Resistance Integrative Research Group with funding awarded to RS and EEO. SK receives salary support through a Transition Award (MOH-TA19may-0003), and JGL and EEO through a Clinician Scientist Award (MOH-CSAINV19may-0002 and MOH-000332-00 respectively), all awarded by the National Medical Research Council of Singapore. The funders had no role in study design, data collection and analysis, decision to publish, or preparation of the manuscript.

**Competing interests:** The authors have declared that no competing interests exist.

## Author summary

Chronic sequelae after viral infections such dengue have been observed, but their true prevalence and impact remain undefined. We prospectively enrolled a cohort of 209 patients with dengue and acute viral respiratory infections (ARI) and followed them up chronic sequelae for up to one year. 18% of patients in the dengue cohort and 14.6% of patients in the ARI cohort experienced chronic sequelae such as fatigue, somnolence, headache, concentration impairment and memory impairment. Patients who experienced chronic sequelae also had lower month-3 SF-12v2 Mental Component Summary Scores, suggesting that such those who self-reported persistence of symptoms experienced functionally worse mental health. Overall our findings reveal an under-appreciated burden of chronic sequelae in dengue and ARI patients and call for further studies to define the pathophysiology and potential therapeutic options for this condition.

## Introduction

As the Coronavirus Disease 2019 (Covid-19) pandemic continues to affect millions around the world, there has been increasing awareness on the potentially disabling syndrome of long Covid. Long Covid afflicts a significant proportion of patients after convalescence from Severe Acute Respiratory Syndrome Coronavirus 2 (SARS-CoV-2) infection, and includes symptoms such as fatigue, "brain fog" and even depression, which persist for 3 months or more acute infection [1–3]. Yet, this syndrome of post-infectious chronic sequelae is not confined exclusively to SARS-CoV-2, but has also been observed after convalescence from other acute viral infections, one of which is dengue [4].

Dengue is an acute mosquito-borne viral illness that afflicts an estimated one hundred million people annually [5,6]. There is no licensed anti-dengue therapeutic, and the only currently licensed dengue vaccine has safety and efficacy concerns [7]. The acute symptoms of infection usually last 7–10 days, and range from self-limiting undifferentiated fever, to more severe and potentially fatal dengue haemorrhagic fever and shock [5]. Much less however, is understood about post-dengue chronic sequelae. Although symptoms such as fatigue and cognitive impairment have been described post-infection, many of these observations were made retrospectively and thus subject to recall bias, or were derived after a relatively short period of study follow-up [8–13]. As such, the true prevalence and duration of post-dengue chronic sequelae remain poorly defined. Such sequelae would not only have a direct impact on patient convalescence, they would also detrimentally affect the economic productivity of many individuals living in dengue endemic areas. Consequently, the true societal and economic impact of dengue is likely underestimated [14].

To understand the true prevalence of post-dengue chronic sequelae, we designed a Mobile-phone Application for Information extraction in Dengue (MAIDEN). The use of MAIDEN enabled data collection to be conducted remotely, removing the need for frequent in-person study visits, thus minimising study costs and avoiding high participant drop-out rates often associated with large prospective studies. In addition, we also took advantage of MAIDEN to explore the impact of chronic sequelae in patients with acute viral respiratory infections (ARI), another highly prevalent group of acute viral infections prior to the emergence of SARS-CoV-2, but in which the occurrence of long-term sequelae is even more poorly defined. We prospectively enrolled two separate cohorts of patients with confirmed dengue and ARI and followed them for up to one-year post-illness onset. Our findings suggest that chronic sequelae such as

fatigue and central nervous system (CNS)-related symptoms are prevalent in adults after acute dengue, and also affect a sizeable proportion of patients post-ARI.

## Methods

### Ethics statement

The study was approved by the local institutional ethics committee (SingHealth Centralised Institutional Review Board; Approval No: CIRB 2017/2308), and written informed consent was obtained from all participants.

### Study participants

Patients were recruited from the inpatient wards, emergency department and staff healthcare clinic of two hospitals in Singapore, between 20 September 2017 and 29 March 2019.

Patients were divided into two cohorts: acute dengue and those with ARI. Patients were included in the dengue cohort if they had a confirmed diagnosis of acute dengue based on compatible clinical features, positive serum dengue NS1 antigen and/or dengue virus (DENV) polymerase chain reaction (PCR). Patients were included in the ARI cohort if they presented with symptoms compatible with a viral upper respiratory tract infection (fever with rhinorrhea, sore throat or cough) or had a viral respiratory tract infection confirmed by respiratory virus multiplex PCR from a respiratory sample within 48 hours prior to enrolment. All patients enrolled had to have access to a mobile phone with internet capability. In both the dengue and ARI cohorts, patients were excluded if they had active malignancy, neurological, rheumatological or psychiatric conditions, dementia or cognitive impairment, concomitant bacterial infection, pneumonia, surgery within the preceding 3 months, previous hospitalization in the past 6 months, were receiving immunosuppressive therapy, or had chronic viral infection such as human immunodeficiency virus or hepatitis B or C.

### Laboratory methods

Serum dengue NS1 was measured using a commercially available single step EIA (Platelia Dengue NS1 Antigen, BioRad). DENV genome was detected using a multiplex real-time polymerase chain reaction (PCR) (FTD Zika/Dengue/Chik, Fast Track Diagnostics, Luxembourg) on RNA. Respiratory virus multiplex PCR was performed using an in-house protocol based on the Anyplex II RV16 respiratory virus panel (Seegne Inc.), and included the following viruses: respiratory syncytial viruses A and B, influenza A and B, parainfluenza viruses 1–4, metapneumovirus, rhinovirus, human coronavirus OC43, 229E and NL63, adenovirus, human enterovirus and human bocavirus 1–4.

### Clinical data collection

At enrolment after informed consent, relevant demographic and clinical data were collected from patients' medical records. Patients were sent a web link via e-mail to download the MAIDEN application onto their mobile phone, as well as a unique user-ID and password to enable them to access the application. A member of the study team then provided a short tutorial on how to use MAIDEN (Fig 1A). At each visit, patients would enter information regarding symptoms experienced (including the absence or presence of fatigue, somnolence, headache, concentration impairment and/or memory impairment) and self-reported overall health into MAIDEN (see Supporting Information S1 Study Questionnaire for the full list of questions). The overall health assessments were adapted from the SF-12v2 health survey, which is a validated tool for self-reported physical and mental wellbeing measurement [15–

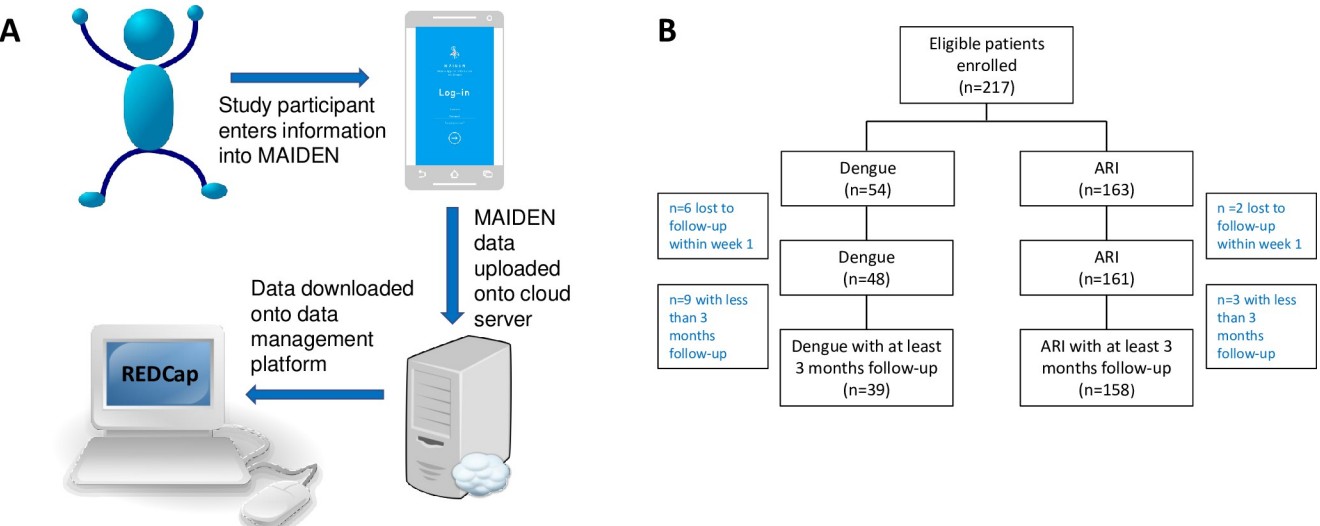

**Fig 1. Study design and enrolment** (A) Study data was collected using the Mobile Application for Information collection in Dengue (MAIDEN) application. At each study visit, patients would enter information remotely into MAIDEN which was installed on their mobile phone. The data collected would then be uploaded onto a secure cloud server and downloaded into the RedCap data management platform where it could be accessed by the study team. (B) Consort flow of patients enrolled onto the study. Patients were excluded from the overall analysis if they dropped out of the study or were lost-to-follow up within the first week after enrolment.

18]. Briefly, the SF-12v2 consists of 12 questions that measure eight health domains to assess physical and mental health. Specifically, mental health-related scales include Vitality (VT), Social Functioning (SF), Role Emotional (RE), and Mental Health (MH). Mental Composite Summary Scores (MCS) were calculated using the weighted means of each mental health domain, with a score of 50 considered the population average. Thus, a score below 50 would indicate worse mental health than the expected average, and a score above 50 would indicate better mental health.

Patients were followed-up daily for the first 7 days, weekly from weeks 2–8, and then monthly from month-3 onwards, up to one-year following acute illness. Except for the first visit at study enrolment, patients were not required to attend the study site in-person, and would be sent reminders by either short messaging system (SMS) or push notification to enter data remotely into MAIDEN when each study "visit" was due. If patients encountered any technical difficulties or had questions about the study, they could contact the study team directly either via phone call or email, or through a direct messaging system within MAIDEN. All data entered into MAIDEN was stored on a secure cloud server and downloaded onto a data management platform for analysis.

## Definitions

We defined "chronic sequelae" as symptoms (fatigue, somnolence, headache, concentration impairment or memory impairment) lasting for at least 3-months post-infection. A patient's symptoms were considered resolved if they did not report any symptoms for at least 2-months after the last date of symptoms reported, and were considered lost to follow-up if they did not enter any data into MAIDEN for more than a 2-month period.

## Statistical analysis

Differences in demographic features, co-morbidities, prevalence of symptoms at enrolment and MCS between the dengue and ARI cohorts, and those with and without chronic sequelae

were analysed using the Fisher's exact test for categorical variables or Mann-Whitney test for continuous variables. Time to resolution of symptoms in both the dengue and ARI cohorts were analysed using Kaplan-Meier analyses. The SF12v2 MCS was calculated using the Optum PRO CoRE software. P-values <0.05 were considered statistically significant. All statistical analyses were performed using GraphPad Prism version 9.2.0.

## Results

A total of 209 patients were enrolled during the study period, with 48 patients in the dengue cohort, and 161 patients in the ARI cohort. 81.3% (39/48) of dengue patients and 98.1% (158/161) of ARI patients completed at least 3 months of study follow-up (Fig 1B). Table 1 shows the demographic and clinical profile of both patient cohorts. Baseline demographics such as age, ethnicity and co-morbidities were balanced between both groups, except for gender, with more males in the dengue cohort (63% vs. 29%, p<0.001). More patients in the dengue cohort were hospitalized compared to those in the ARI cohort (77.1% vs. 18.1%, p<0.001), reflecting differences in clinical management between these two disease types. Only one patient had severe dengue, as defined by the World Health Organization (WHO) 2009 dengue classification scheme [19]. Within the ARI cohort, the specific viral etiology was tested for in 29 patients. Among these patients, influenza virus and rhinovirus were the most common etiologies identified (Table 1).

Almost all patients (95.8% of dengue and 94.4% of ARI patients) experienced fatigue or at least one other CNS-related symptom (somnolence, headache, concentration impairment and/or memory impairment) within the first week after enrolment. Fatigue and somnolence were the most commonly reported symptoms in the dengue cohort, while lethargy and headache were most common in the ARI cohort (Table 1). Amongst patients with at least 3-months of follow-up, 18.0% in the dengue cohort and 14.6% in the ARI cohort experienced chronic sequelae (Table 2). In the dengue cohort, fatigue, concentration impairment and memory impairment were the most commonly reported chronic symptoms, while fatigue and headache were most prevalent in the ARI cohort. Kaplan-Meier analysis indicated that fatigue and CNS-related symptoms resolved within weeks of convalescence in the majority of dengue patients. However, the rate of full recovery then plateaued and even persisted in some patients until the end of our observation period (Fig 2A). Similar trends were observed in patients with ARI (Fig 2B). No statistically significant difference in age, gender distribution or hospitalisation status was observed between those with and without chronic sequelae (Table 3).

In order to determine if self-reported symptoms of chronic sequelae translated into worse mental health outcomes, we compared the SF12v12 MCS scores at month 3 in patients with and without chronic sequelae. In the dengue and ARI cohorts combined, the median month 3 MCS was lower in patients who remained symptomatic at 3 months and beyond, compared to those whose symptoms fully resolved (47.7 vs. 56.0, p<0.001) (Fig 3A), indicating that patients who self-reported persistence of symptoms beyond 3 months also experienced functionally worse mental health. This difference in the median month 3 MCS scores between those with and without chronic sequelae was also observed in the ARI cohort alone (46.6 vs. 56.0, p<0.01) (Fig 3B). Although a similar trend was observed in the dengue cohort, this did not reach statistical significance, likely due to the smaller sample size in this cohort (50.0 vs. 55.7, p = 0.20) (Fig 3C).

Overall, our findings indicate that a significant proportion of both dengue and ARI patients experienced chronic sequelae, and the persistence of such chronic sequelae translated to overall worse mental health outcomes.

**Table 1. Demographics and clinical features of the study cohort.** *Within the ARI cohort, the specific viral etiology was known in 29 patients. The percentage reported is out of a total of 29 patients. Of these 29 patients, 4 patients tested positive for two different viruses concurrently on multiplex PCR–influenza and adenovirus (n = 1), influenza and human coronavirus OC43 (n = 1), parainfluenza and respiratory syncytial virus (n = 1), and parainfluenza and enterovirus virus (n = 1). **Any one of fatigue, somnolence, headache, concentration impairment or memory impairment.

| | Dengue Cohort (n = 48) | ARI Cohort (n = 161) | p-value |
|---|---|---|---|
| **Demographics** | | | |
| Age | | | |
| Median (range), years | 37.0 (21–68) | 34.0 (22–79) | 0.39 |
| Male sex, no. (%) | 30 (62.5) | 46 (28.6) | **<0.001** |
| Ethnicity, no. (%) | | | 0.21 |
| Chinese | 29 (60.4) | 77 (47.8) | |
| Malay | 12 (25.0) | 36 (22.4) | |
| Indian | 3 (6.3) | 21 (13.0) | |
| Others | 4 (33.3) | 27 (16.8) | |
| Co-morbidities, no. (%) | | | |
| Diabetes mellitus | 2 (4.2) | 5 (3.1) | 0.72 |
| Hypertension | 4 (8.3) | 16 (9.9) | 0.74 |
| Ischemic heart disease | 2 (4.2) | 2 (1.2) | 0.19 |
| **Disposition** | | | |
| Hospitalised, no (%) | 37 (77.1) | 29 (18.0) | **<0.001** |
| **Study Compliance** | | | |
| Completed at least 3 months of study follow-up, no (%) | 39 (81.3) | 158 (98.1) | **<0.001** |
| **Viral Etiology** | | | |
| Severe dengue, no. (%) | 1 (2.0) | - | NA |
| Respiratory virus type, no. (%)* | | | |
| Influenza A & B | - | 14 (48.3) | |
| Rhinovirus | - | 6 (20.7) | |
| Adenovirus | - | 4 (13.8) | |
| Human coronavirus (OC43) | - | 4 (13.8) | |
| Parainfluenza | - | 3 (10.3) | |
| Human coronavirus (NL63) | - | 1 (3.4) | |
| Enterovirus | - | 1 (3.4) | |
| **Clinical Features** | | | |
| Symptom type at presentation, no (%) | | | |
| Any symptom** | 46 (95.8) | 152 (94.4) | 1.00 |
| Fatigue | 38 (79.2) | 136 (84.5) | 0.51 |
| Somnolence | 32 (66.7) | 93 (57.8) | 0.32 |
| Headache | 29 (60.4) | 119 (74.0) | 0.07 |
| Concentration impairment | 29 (60.4) | 95 (59.0) | 0.86 |
| Memory impairment | 13 (27.1) | 28 (17.4) | 0.15 |

## Discussion

To date, few studies have examined the long-terms health effects of either acute dengue or non-SARS-CoV-2 ARI, making it difficult to quantify the true societal and economic impact of such acute viral infections. In this prospective study of over 200 patients, we found that a significant proportion of dengue, and even more unexpectedly ARI patients, continue to experience fatigue and CNS-related chronic sequelae for several months after resolution of the acute infection, not dissimilar to long Covid sufferers [1,2].

**Table 2. Symptoms in patients with chronic sequelae.** Analysis performed on patients with at least 3 months of study follow-up (dengue: n = 39, ARI: n = 158). **Any one of fatigue, somnolence, headache, concentration impairment or memory impairment.

|  | Dengue Cohort (n = 39) | ARI Cohort (n = 158) | p-value |
|---|---|---|---|
| **Chronic Sequelae** |  |  |  |
| Any symptom persisting ≥ 3 months*, no. (%) | 7 (18.0) | 23 (14.5) | 0.62 |
| Symptom type at month 3, no (%) |  |  |  |
| Fatigue | 4 (10.3) | 15 (9.5) | 1.00 |
| Somnolence | 1 (2.6) | 10 (6.3) | 0.47 |
| Headache | 2 (5.1) | 13 (8.2) | 0.74 |
| Concentration impairment | 4 (10.3) | 10 (6.3) | 0.48 |
| Memory impairment | 4 (10.3) | 8 (5.1) | 0.25 |

At present, it still remains unclear what the drivers of such chronic sequelae are, and unlike Covid-19, little research has focused on the mechanistic causes of chronic sequelae post-dengue. This is even less so for acute respiratory viral infections such as influenza and adenovirus. In one study of dengue patients, an association was found between the FcγRIIa (FcγRIIa-131HH) gene polymorphism, the presence of autoimmune markers and symptom persistence [9]. Apart from this report, no other studies to date explored the causes of post-dengue chronic sequelae.

It is possible that aberrant activation of the innate immune response to the original viral stimuli may result in chronic inflammation and subsequent long term tissue damage and immune exhaustion, which may then in turn drive the development of chronic sequelae. Indeed, we have recently shown that increased baseline expression of genes associated with T-cell exhaustion was associated with the development of fatigue after SARS-CoV-2 mRNA vaccination [20]; such immune dysregulation has also been associated with the development of chronic fatigue syndrome/myalgic encephalomyelitis [21–23]. In the context of long Covid,

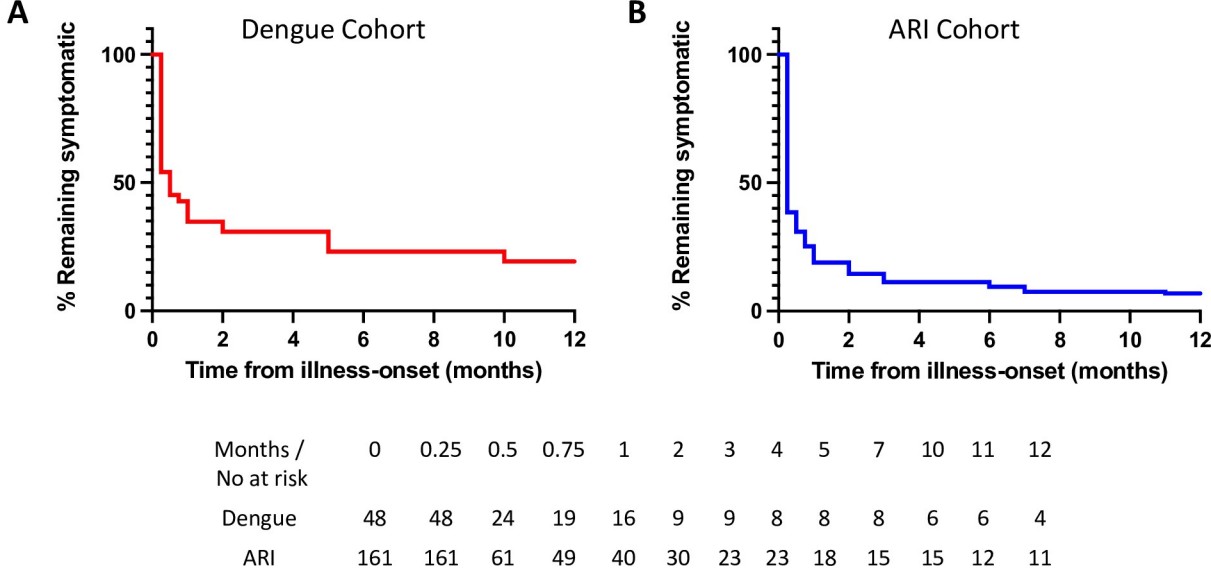

**Fig 2. Kaplan-Meier curve of time to symptom resolution.** Each curve shows the overall time to symptom resolution (in months) after illness-onset in patients with (A) dengue and (B) acute viral respiratory infection (ARI). The numbers of patients at risk for each of these two groups are shown in the table immediately below the Kaplan-Meier curves.

**Table 3. Demographics of patients with and without chronic sequelae.** Chronic sequelae is defined as persistence of symptoms (any one of fatigue, somnolence, headache, concentration impairment or memory impairment) for at least 3 months.

| | Dengue Cohort (n = 39) | | | ARI Cohort (n = 158) | | | All (n = 197) | | |
|---|---|---|---|---|---|---|---|---|---|
| | Chronic sequelae | | | Chronic sequelae | | | Chronic sequelae | | |
| | Absent (n = 32) | Present (n = 7) | p-value | Absent (n = 135) | Present (n = 23) | p-value | Absent (n = 167) | Present (n = 30) | p-value |
| **Age, years** | | | | | | | | | |
| Median (range) | 38.5 (21–68) | 51 (24–60) | 0.50 | 35.0 (23–67) | 31.0 (22–64) | 0.27 | 35.0 (21–68) | 31.5 (22–64) | 0.72 |
| **Gender** | | | | | | | | | |
| Male sex, no. (%) | 21 (65.6) | 3 (42.9) | 0.40 | 40 (29.6) | 6 (26.1) | 0.81 | 61 (36.5) | 9 (30.0) | 0.54 |
| **Disposition** | | | | | | | | | |
| Hospitalised, no. (%) | 26 (81.3) | 6 (85.7) | 1.00 | 20 (14.8) | 7 (30.4) | 0.08 | 46 (27.5) | 13 (43.3) | 0.09 |

multiple mechanisms have been postulated, including chronic inflammation, antigen persistence, latent virus reactivation and development of auto-antibodies [2–4,24]. It is likely that some, if not all of these mechanisms may also apply to the development of chronic sequelae after other acute viral infections. The prevalence of such chronic sequelae and their functional impact on mental well-being underscore the need for more in-depth research to understand the specific drivers of such post-infection chronic outcomes.

We used MAIDEN as a study tool for remote data-collection in order to overcome the high financial costs and participant drop-out rates often associated with large prospective studies. We reasoned that remote data collection would promote better compliance from study participants in view of the convenience afforded through remote data collection. Indeed, overall study compliance was good with over 90% of patients completing at least 3 months of study follow-up. Besides improving study visit compliance, mobile data collection also allows for

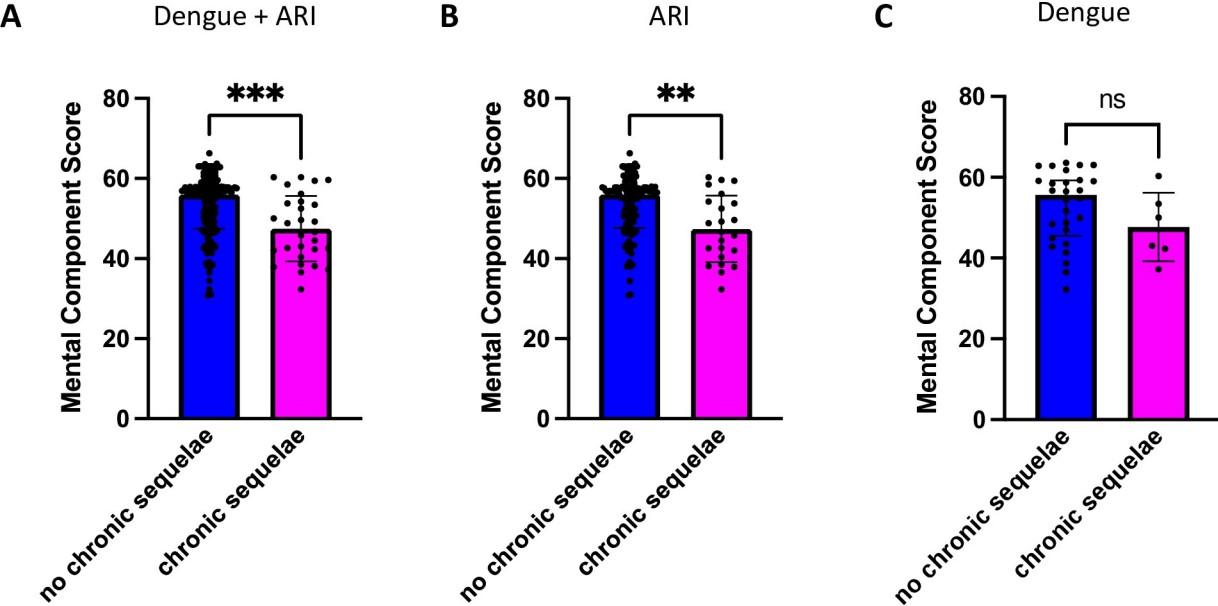

**Fig 3. SF12v2 Mental Component Summary (MCS) Score at month 3 in patients with and without chronic sequelae.** (A) Comparison of MCS scores in both the dengue and ARI cohorts combined. (B) Comparison of MCS scores within the ARI cohort. (C) Comparison of MCS scores within the dengue cohort. Blue bars represent the group without chronic symptoms while magenta bars represent the group with chronic symptoms. Black dots indicate the individual MCS values. Summary plots show the median MCS with error bars representing the interquartile range. Statistics to compare the median MCS between groups was calculated using the Mann-Whitney test. *** = p < 0.001, **** = p <0.0001, ns = not significant.

immediate pooling of data into a centralized data warehouse, making real-time data analysis possible. Given these benefits, and the fact that over 80% of the world's population own a smartphone [25], we envisage that mobile phone applications such as MAIDEN could be a useful adjunct tool not only in the conduct of observational cohort studies, but also in the clinical trial space where such applications could be used for adverse event tracking and reporting.

A strength of our study is its prospective design, which avoids the problem of recall bias often associated with retrospective studies. The study follow-up period of one year also enabled the tracking of patients longitudinally over time, allowing for clearer evaluation of time to symptom resolution. In our analysis, we did not include symptoms that resolved and then re-occurred after more than a two-month period in order to minimize the possibility that such symptoms were due to other acute illnesses that developed during the period of observation. Although the symptoms reported by patients were subjective in nature, the use of the validated SF12v2 health outcomes questionnaire allowed for a functional assessment of each study participant's overall mental health. We showed that patients who reported persistence of fatigue and/or CNS-related symptoms beyond 3 months had a significantly lower MCS then their counterparts whose symptoms lasted less than 3 months, indicating that self-reported chronic sequelae translates into functionally worse mental health outcomes. Indeed, structural brain changes on magnetic resonance imaging have been found in patients with post-dengue neurological sequelae such as encephalopathy, suggesting a pathological basis for such sequelae [26].

A limitation of our study is that patients in the ARI cohort were not tested specifically for DENV, and hence we are unable to completely exclude co-infection. We also acknowledge that the hospitalisation status was imbalanced between the two study cohorts, with a significantly higher proportion of the dengue cohort hospitalised at the time of enrolment, compared to the ARI cohort where the majority of patients were ambulatory. As such, we avoided making head-to head comparisons between the two cohorts. Finally, we acknowledge that our study sample size was relatively small, particularly in the dengue cohort, as case numbers in Singapore during the study enrolment period were low. Nevertheless, we believe that our prospective study and data collection methods have generated reasonable evidence of post-infective chronic sequelae in our study cohort.

In conclusion, we show that persistent fatigue and CNS-related chronic sequelae, as well as poorer overall mental health, occur in a significant proportion of both dengue and ARI patients post-infection. Our findings reveal an oft under-appreciated burden of post-infection chronic sequelae in both dengue and ARI patients, and highlight the need for therapeutic and preventative strategies in order to prevent both acute viral infection and its associated chronic sequelae.

## Supporting information

**S1 Study Questionnaire. MAIDEN content and questions for study participants.** (DOCX)

## Acknowledgments

We thank the clinical research coordinators from the Department of Infectious Diseases, Singapore General Hospital and the Clinical Trials & Research Unit, Changi General Hospital for assisting with this study, and the patients who volunteered for the study.

## Author Contributions

**Conceptualization:** Shirin Kalimuddin, Shay Paintal, Ram Sasisekharan, Jenny G. Low, Eng Eong Ooi.

**Formal analysis:** Shirin Kalimuddin, Liang En Wee, Eng Eong Ooi.

**Funding acquisition:** Yii Ean Teh, Ram Sasisekharan.

**Investigation:** Shirin Kalimuddin, Yii Ean Teh, Jenny G. Low, Sujata K. Sheth.

**Methodology:** Shirin Kalimuddin, Shay Paintal, Ram Sasisekharan, Eng Eong Ooi.

**Software:** Shay Paintal.

**Supervision:** Shirin Kalimuddin, Sujata K. Sheth.

**Visualization:** Yii Ean Teh.

**Writing – original draft:** Shirin Kalimuddin.

**Writing – review & editing:** Shirin Kalimuddin, Yii Ean Teh, Liang En Wee, Jenny G. Low, Eng Eong Ooi.

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
