## [Decision Letter · Decision Letter 0]

5 Jul 2022

Dear Dr Kalimuddin,

Thank you very much for submitting your manuscript "Chronic sequelae complicate convalescence from both dengue and acute viral respiratory illness" for consideration at PLOS Neglected Tropical Diseases. As with all papers reviewed by the journal, your manuscript was reviewed by members of the editorial board and by several independent reviewers. The reviewers appreciated the attention to an important topic. Based on the reviews, we are likely to accept this manuscript for publication, providing that you modify the manuscript according to the review recommendations. 

Sincerely,

Rhoel Ramos Dinglasan

Associate Editor

Samuel Scarpino

Deputy Editor

Reviewer's Responses to Questions

**Key Review Criteria Required for Acceptance?**

**Methods**

-Are the objectives of the study clearly articulated with a clear testable hypothesis stated?

-Is the study design appropriate to address the stated objectives?

-Is the population clearly described and appropriate for the hypothesis being tested?

-Is the sample size sufficient to ensure adequate power to address the hypothesis being tested?

-Were correct statistical analysis used to support conclusions?

-Are there concerns about ethical or regulatory requirements being met?

Reviewer #1: The manuscript discussed the post-infection long sequelae, on both dengue and acute respiratory infection (ARI), to evaluate the post-infection symptoms and its impact on patients' mental health. The study is clearly defined, with appropriate objective and hypothesis. The study design is suitable to answer the research question. The population is sufficient, with limitation has already been mentioned and acknowledged by the authors. The sample size might be underwhelming, as the prevalence of both diseases are massive in Singapore, however, I would believe it is still acceptable. The statistical analysis was also appropriate.

Overall, I would deem the study is well-written, with appropriate design.

Reviewer #2: This study has been well planned, with validated and well characterized study tools used and patients have been prospectively followed up. Symptoms have been recorded using a mobile app, so that symptoms are not missed. However, the main limitation of this study is the small sample size of patients with dengue (n=48), with only one patient having severe dengue. While the results highlight some important findings, it would be very interesting to see the results in a larger sample size. This study has been compared with the chronic sequel of COVID-19. In such studies, the sample sizes have been extremely large. Therefore, it would be important to see these results in a larger study population as well, in patients with varying clinical disease severity. It would be best if the authors discuss the limitations in their sample size in the discussion. 

There are no concerns regarding ethical or regulatory requirements.

Reviewer #3: Minor revision

How could you distinguished between symptom occurred at 3 months or during monitoring were belongs to dengue not belongs to other acute symptoms or conditions?

Authors also included patients with co-morbidities, those co-morbidities can contribute to symptoms during monitoring.

Do you have records of any acute symptoms (for example, another dengue infection, another ARI) during monitoring i.e fever, diarrhoea, cough, sneezing, myalgia. If you had, did you do laboratory testing to find out the possible cause of aetiology?

**Results**

-Does the analysis presented match the analysis plan?

-Are the results clearly and completely presented?

-Are the figures (Tables, Images) of sufficient quality for clarity?

Reviewer #1: I can't find the use of SF12v2 result on the result section; however, it seems to be used on Figure 3, i would suggest adding more elaborate explanation on this. Thus, the readers would be clear on the comparison of both populations. More elaborate discussion on this topic would also be more appreciated.

Reviewer #2: The study is well planned and executed, with appropriate data analysis plan. However, the sample size is quite small.

Reviewer #3: The analysis presented match the analysis plan.

The results are clearly presented

**Conclusions**

-Are the conclusions supported by the data presented?

-Are the limitations of analysis clearly described?

-Do the authors discuss how these data can be helpful to advance our understanding of the topic under study?

-Is public health relevance addressed?

Reviewer #1: The conclusions are well presented, and limitation has already been acknowledged by the authors. The data can be used to properly evaluate the post infection sequelae that might be found in both dengue and ARI patients, to support whole comprehensive treatment. The relevance to public health is highly correlated.

Reviewer #2: The authors state that ‘Apart from this report, no other studies to date explored the causes of post dengue chronic sequelae’ : However, this statement is not very accurate as there have been studies looking at chronic sequel, although not using the same study tools as described in this study. It would be important to discuss the following studies and how these previous studies relate to this study. 

• 2 month follow up study of 177 patients: https://pubmed.ncbi.nlm.nih.gov/17137834/

• Cost of persistent dengue: Persistent Symptoms of Dengue: Estimates of the Incremental Disease and Economic Burden in Mexico

• 2 month follow up study of 158 patients: https://pubmed.ncbi.nlm.nih.gov/33099653/

The authors discuss the possible causes of chronic sequel of dengue. It would be also important to discuss long term autoimmune neurological diseases and other autoimmune diseases following dengue

 Pattern Recognition Approach to Brain MRI Findings in Patients with Dengue Fever with Neurological Complications: https://pubmed.ncbi.nlm.nih.gov/33109849/

Reviewer #3: Authors should described more about limitation that study could not absolutely sure that the symptoms occurred during monitoring may belongs to other disease/condition

**Editorial and Data Presentation Modifications?**

Reviewer #1: Data of MSC or SF12v2 should be added to the manuscript. Otherwise, the data is well presented.

Reviewer #2: (No Response)

Reviewer #3: Minor revision

**Summary and General Comments**

Reviewer #1: The study is well-designed and well-written. Statistical analysis and research design were appropriate for this study. The study also addressed the highly-neglected impact of post-infection sequelae in dengue and ARI patients. Overall, i would deem the study to only need minor-revision.

Reviewer #2: This is an important study, characterizing the chronic complications of dengue, which is very much neglected. However, the sample size is very small, which is the main limitation. Furthermore, although the authors state that this is the only study that has been carried out, there have been other studies, with larger sample size, but shorter follow up period being carried out.

Reviewer #3: This is an important study that should be done in order to know whether dengue really has chronic sequelae or not.

PLOS authors have the option to publish the peer review history of their article (what does this mean?). If published, this will include your full peer review and any attached files.

Reviewer #1: Yes: Lowilius Wiyono

Reviewer #2: No

Reviewer #3: No

Figure Files:

Data Requirements:

Reproducibility:

References

---

## [Decision Letter · Decision Letter 1]

8 Aug 2022

Dear Dr Kalimuddin,

We are pleased to inform you that your manuscript 'Chronic sequelae complicate convalescence from both dengue and acute viral respiratory illness' has been provisionally accepted for publication in PLOS Neglected Tropical Diseases.

Best regards,

Rhoel Ramos Dinglasan

Academic Editor

Samuel Scarpino

Section Editor

Reviewer's Responses to Questions

**Key Review Criteria Required for Acceptance?**

**Methods**

-Are the objectives of the study clearly articulated with a clear testable hypothesis stated?

-Is the study design appropriate to address the stated objectives?

-Is the population clearly described and appropriate for the hypothesis being tested?

-Is the sample size sufficient to ensure adequate power to address the hypothesis being tested?

-Were correct statistical analysis used to support conclusions?

-Are there concerns about ethical or regulatory requirements being met?

Reviewer #1: The study has already stated clear objective and hypothesis, which enforce the need to see the impact of longstanding ARI and dengue with its chronic sequelae. The study is designed appropriately with appropriate population. I believe the authors have used correct statistical analysis.

Reviewer #2: The authors have addressed all my questions and I believe is suitable for publication.

Reviewer #3: (No Response)

**Results**

-Does the analysis presented match the analysis plan?

-Are the results clearly and completely presented?

-Are the figures (Tables, Images) of sufficient quality for clarity?

Reviewer #1: The analysis is well-matched with the methodology mentioned. The results are clear and easy to understand. I believe the authors have revised the manuscript well and incorporate the results and discussion in a very comprehensive way

Reviewer #2: As mentioned earlier, the limitation of the study is the sample size. The authors have added this as a limitation of the study.

Reviewer #3: (No Response)

**Conclusions**

-Are the conclusions supported by the data presented?

-Are the limitations of analysis clearly described?

-Do the authors discuss how these data can be helpful to advance our understanding of the topic under study?

-Is public health relevance addressed?

Reviewer #1: The conclusion is clearly presented with the supporting data and enforce the need of further supervision and management on these patients. I believe the conclusion are also important to be noticed in the management of these patients.

Reviewer #2: (No Response)

Reviewer #3: (No Response)

**Editorial and Data Presentation Modifications?**

Reviewer #1: No revision needed

Reviewer #2: (No Response)

Reviewer #3: (No Response)

**Summary and General Comments**

Reviewer #1: The study is well-designed with comprehensive discussion on the topic. The authors have addressed their limitation to be objectively discussed by the readers. As a reviewer, I would believe this manuscript suffice to be published.

Reviewer #2: (No Response)

Reviewer #3: (No Response)

PLOS authors have the option to publish the peer review history of their article (what does this mean?). If published, this will include your full peer review and any attached files.

Reviewer #1: **Yes: **Lowilius Wiyono

Reviewer #2: No

Reviewer #3: No

---

## [Editor Report · Acceptance letter]

15 Aug 2022

Dear Dr Kalimuddin,

We are delighted to inform you that your manuscript, "Chronic sequelae complicate convalescence from both dengue and acute viral respiratory illness," has been formally accepted for publication in PLOS Neglected Tropical Diseases.

Best regards,

Shaden Kamhawi

co-Editor-in-Chief

Paul Brindley

co-Editor-in-Chief
